# Antidepressant-like Effects of *Cannabis sativa* L. Extract in an Lipopolysaccharide Model: Modulation of Mast Cell Activation in Deep Cervical Lymph Nodes and Dura Mater

**DOI:** 10.3390/ph17101409

**Published:** 2024-10-21

**Authors:** Joonyoung Shin, Dong-Uk Kim, Gi-Sang Bae, Ji-Ye Han, Do-Won Lim, Young-Mi Lee, Eunjae Kim, Eunjeong Kwon, Dongwoon Han, Sungchul Kim

**Affiliations:** 1Institute for Global Rare Disease Network, Professional Graduate School of Korean Medicine, Wonkwang University, Iksan 54538, Republic of Korea; spm1219@naver.com (J.S.); dwhan@hanyang.ac.kr (D.H.); 2Department of Pharmacology, School of Korean Medicine, Wonkwang University, Iksan 54538, Republic of Korea; ckck202@naver.com (D.-U.K.); baegs888@wku.ac.kr (G.-S.B.); 3Department of Oriental Pharmacy, Wonkwang-Oriental Medicines Research Institute, College of Pharmacy, Wonkwang University, Iksan 54538, Republic of Korea; hsue0112@gmail.com (J.-Y.H.); limsc05@wku.ac.kr (D.-W.L.); ymlee@wku.ac.kr (Y.-M.L.); 4College of Pharmacy, Kyung Hee University, Seoul 02447, Republic of Korea; dmswo3537@naver.com (E.K.); dmswjd4876@naver.com (E.K.); 5Department of Global Health and Development, Hanyang University, Seoul 04763, Republic of Korea

**Keywords:** *Cannabis sativa* L. inflorescence extract, lipopolysaccharide, mast cells, cytokine, deep cervical lymph nodes, dura mater

## Abstract

Background: Lipopolysaccharide (LPS)-induced neuroinflammation is a well-established model for studying depression-like behavior, driven by pro-inflammatory cytokines such as TNF-α and IL-1β. Mast cells (MCs) contribute to neuroinflammation by releasing mediators that exacerbate depressive-like symptoms. This study evaluates the antidepressant-like and anti-inflammatory effects of *Cannabis sativa* L. inflorescence extract (CSL) in an LPS-induced neuroinflammation model. Methods: Male C57BL/6 mice were intraperitoneally injected with CSL at doses of 10, 20, and 30 mg/kg, 30 min prior to LPS (0.83 mg/kg) administration. Depressive behaviors were assessed using the sucrose preference test (SPT), tail suspension test (TST), and forced swimming test (FST). The neutrophil-to-lymphocyte ratio (NLR) was measured to assess systemic inflammation. Cytokine levels in the prefrontal cortex (PFC) were measured, and mast cell degranulation in the lymph nodes and dura mater was analyzed histologically (approval number: WKU24-64). Results: CSL significantly improved depressive-like behaviors and decreased the NLR, indicating reduced systemic inflammation. CSL also significantly reduced TNF-α and IL-1β levels in the PFC. Furthermore, CSL inhibited MC degranulation in the deep cervical lymph nodes and dura mater, with the strongest effects observed at 30 mg/kg. Conclusions: CSL demonstrated antidepressant-like and anti-inflammatory effects in an LPS-induced neuroinflammation model, likely through the modulation of cytokine expression and mast cell activity. These results suggest the potential of CSL as a therapeutic option for treating inflammation-related depression.

## 1. Introduction

Major Depressive Disorder (MDD) is a widespread and severe mental health condition, characterized by persistent symptoms. These include profound sadness, loss of interest in previously pleasurable activities (anhedonia), motor agitation or slowing, feelings of guilt or worthlessness, difficulty focusing, and recurring thoughts of death or suicide [1]. Recent studies are exploring the connection between elevated levels of pro-inflammatory cytokines, including tumor necrosis factor-alpha (TNF-α), Interleukin 1 beta (IL-1β), and IL-6, and the development of MDD [2,3]. Mast cells (MCs), key components of the innate immune system, have been implicated in these neuroinflammatory processes by releasing cytokines, chemokines, and other pro-inflammatory mediators [4]. These immune activities are thought to contribute to psychiatric disorders like MDD by interacting with the brain’s immune system, potentially triggering neuroinflammatory responses that lead to depressive symptoms [5,6,7]. A well-known trigger of immune responses is lipopolysaccharide (LPS), a component of bacterial cell walls. Studies have shown that administering LPS can lead to sickness behaviors in animal models, such as depression-like behavior, anhedonia, and increased anxiety [8,9]. Similarly, LPS has also been associated with depressive symptoms in humans [10,11]. Notably, mice administered LPS (5 mg/kg, i.p.) exhibited dysfunction in deep cervical lymph nodes (dcLNs) and alterations in meningeal lymphatic drainage and morphology, leading to heightened neuroinflammation and cognitive impairment [12]. These findings suggest a critical role for immune system dysfunction in the development and exacerbation of depressive symptoms in MDD.

*Cannabis sativa* L. is a dioecious annual plant known for its diverse pharmacological properties. Cultivated primarily in Central Asia, particularly in regions like India and China, since ancient times [13], *Cannabis sativa* L. contains bioactive compounds, including cannabinoids such as cannabidiol (CBD) and delta-9-tetrahydrocannabinol (Δ9-THC) [14]. These cannabinoids are especially significant because of their therapeutic potential. Δ9-THC, a psychoactive compound, and CBD, a non-psychoactive compound, both interact with endocannabinoid receptors [15], offering therapeutic effects for various conditions, including epilepsy, inflammation, and MDD [16,17,18]. Additionally, cannabidiolic acid methyl ester, a derivative of cannabidiolic acid (CBDA), has demonstrated the ability to reduce stress-induced anxiety-like behaviors in depressed rat models [19]. In addition to cannabinoids, *Cannabis sativa* L. is rich in terpenes, which play a significant role in its wide-ranging pharmacological effects, including anti-inflammatory and pain-relieving properties [20]. Terpenes are among the most varied plant compounds, with key constituents such as α-pinene, β-pinene, myrcene, limonene, β-caryophyllene, and α-humulene. These terpenes offer a wide range of health benefits, including anti-inflammatory, anticancer, and antibacterial effects [21,22]. Interestingly, myrcene has been shown to reduce the expression of pro-inflammatory cytokines such as IL-1β, IL-6, IL-8, and TNF-α [23]. In addition, α-pinene reduces Parkinson’s disease symptoms [24], and alleviates postpartum depression through its antioxidant properties [25]. These findings highlight the diverse bioactive compounds in *Cannabis sativa* L. and their potential role in managing inflammation.

This study aimed to evaluate the effects of *Cannabis sativa* L. inflorescence extract (CSL) on LPS-induced depressive-like behaviors in mice. Additionally, the study assessed the ability of CSL to modulate MC activation in the dcLNs and dura mater, regions known to be involved in neuroinflammation, as well as its impact on the neutrophil-to-lymphocyte ratio (NLR) as an inflammatory marker. Furthermore, we examined the inhibitory effect of CSL on pro-inflammatory cytokines (TNF-α, and IL-1β) in the prefrontal cortex (PFC). These findings suggest that CSL may offer valuable insights into the interactions between mast cell activation and inflammation in neuroinflammatory conditions.

## 2. Results

### 2.1. Cannabinoids and Terpenes Contents in CSL

A typical total ion chromatogram (TIC) for the conversion products of CSL under acidic reaction conditions, as analyzed in gas chromatography–mass spectrometry (GC-MS) scan mode, is shown in Figure 1a. CBDA (peak 6, 1209.39 μg/mg), and CBCA (peak 8, 18.99 μg/mg) were detected as major cannabinoid components of CSL (Figure 1a). Terpenes were detected in CSL using headspace–gas chromatography/mass spectrometry (HS-GC/MS). Myrcene (peak 3, 2284.17 µg/mg) and α-Pinene (peak 1, 1115.25 µg/mg) were detected as major terpenes of CSL (Figure 1b).

### 2.2. Antidepressant-like Effect of CSL in the LPS Model

In the sucrose preference test (SPT), the LPS group presented a decreased sucrose preference (45.35 ± 4.30%) compared to the control (CON) group (77.71 ± 6.86%, *p* < 0.001, Figure 2a). CSL administration significantly increased sucrose preference in a dose-dependent manner compared to the LPS group. The sucrose preference for the 10 mg/kg CSL + LPS group was 58.85 ± 4.58% (*p* < 0.05), for the 20 mg/kg CSL + LPS group was 63.96 ± 10.65% (*p* < 0.001), and for the 30 mg/kg CSL + LPS group was 67.86 ± 5.55% (*p* < 0.001). No significant differences were observed between the CSL + LPS groups. 

In the tail suspension test (TST), the LPS group presented an increased immobility time (220.38 ± 11.84 s) compared to the CON group (82.81 ± 29.56 s, *p* < 0.001, Figure 2b). CSL administration reduced immobility time across all groups compared to the LPS group, but a significant reduction was observed only in the 20 mg/kg CSL + LPS group. The immobility time for the 10 mg/kg CSL + LPS group was 183.36 ± 32.75 s, for the 20 mg/kg CSL + LPS group was 169.81 ± 29.20 s (*p* < 0.05), and for the 30 mg/kg CSL + LPS group was 171.93 ± 35.53 s. No significant differences were observed between the CSL + LPS groups.

In the forced swimming test (FST), the LPS group presented an increased immobility time (152.93 ± 31.68 s) compared to the CON group (52.76 ± 29.53 s, *p* < 0.001, Figure 2c). CSL administration reduced immobility time in a dose-dependent manner compared to the LPS group, but a significant reduction was observed only in the 30 mg/kg CSL + LPS group. The immobility time for the 10 mg/kg CSL group was 115.28 ± 38.77 s, for the 20 mg/kg CSL + LPS group was 101.13 ± 35.91 s, and for the 30 mg/kg CSL + LPS group was 41.56 ± 27.46 s (*p* < 0.001). Notably, the 30 mg/kg CSL + LPS group showed a significantly lower immobility time compared to the other CSL + LPS groups: 10 mg/kg CSL + LPS group (*p* < 0.01) and 20 mg/kg CSL + LPS group (*p* < 0.05).

### 2.3. Hematological Inflammatory Markers: NLR, Monocytes, Eosinophils, and Basophils

The LPS group (2.64 ± 1.25) presented increased neutrophil to lymphocyte ratio (NLR) values compared to the CON group (0.23 ± 0.04, Figure 3a). CSL administration reduced NLR values in a dose-dependent manner compared to the LPS group, with a significant reduction observed in all CSL + LPS groups except the 10 mg/kg CSL + LPS group. The NLR values were 1.81 ± 0.98 for the 10 mg/kg CSL + LPS group, 0.66 ± 0.15 for the 20 mg/kg CSL + LPS group (*p* < 0.001), and 0.52 ± 0.20 for the 30 mg/kg CSL + LPS group (*p* < 0.001). Notably, the 30 mg/kg CSL + LPS group showed significantly lower NLR values compared to the 10 mg/kg CSL + LPS groups (*p* < 0.05, Figure 3a). The LPS group presented increased monocyte (MONO) values (6.00 ± 3.07) compared to the CON group (1.03 ± 0.39, *p* < 0.001, Figure 3b). CSL administration reduced MONO values in a dose-dependent manner compared to the LPS group, but there were no significant differences (Figure 3b). The MONO values were 5.00 ± 2.05 for the 10 mg/kg CSL + LPS group, 4.48 ± 0.95 for the 20 mg/kg CSL + LPS group, and 4.10 ± 1.69 for the 30 mg/kg CSL + LPS group. For eosinophils (EOS) and basophils (BASO), no significant differences in values were observed between each group.

### 2.4. Activation of Mast Cells (MCs) in Lymph Nodes (LNs)

The LPS administration showed no significant differences in the area of lymph nodes (LNs) and the number of MCs in both the dcLNs and submandibular lymph nodes (smLNs) compared to the CON group (Figure 4b,c). CSL administration also did not result in significant differences in LN size or MC counts in either LN type. On the other hand, CSL significantly decreased the degranulation rate of MCs in both node types compared to the LPS group.

In the smLNs, the LPS group exhibited an increased degranulation rate (70.82 ± 5.09%) compared to the CON group (38.01 ± 8.93%, *p* < 0.001, Figure 4d). CSL administration reduced the degranulation rate in a dose-dependent manner, but a significant reduction was observed only in the 30 mg/kg CSL + LPS group. The degranulation rates were 62.51 ± 10.77% for the 10 mg/kg CSL group, 62.71 ± 10.16% for the 20 mg/kg CSL + LPS group, and 51.65 ± 11.53% for the 30 mg/kg CSL + LPS group (*p* < 0.05). 

In the dcLNs, the LPS group showed a higher degranulation rate (66.40 ± 5.43%) compared to the CON group (34.93 ± 6.53%, *p* < 0.001, Figure 4d). CSL reduced the degranulation rate in a dose-dependent manner, with significant reductions observed in all groups except the 10 mg/kg CSL + LPS group. The degranulation rates were 55.16 ± 12.24% for the 10 mg/kg CSL group, 47.60 ± 10.96% for the 20 mg/kg CSL + LPS group (*p* < 0.05), and 44.33 ± 12.81% for the 30 mg/kg CSL + LPS group (*p* < 0.01). Although smLNs generally showed higher values than dcLNs for LN area, MC count, and degranulation rate, these differences were not statistically significant.

### 2.5. Activation of Mast Cells (MCs) in Dura Mater

The LPS administration did not result in significant differences in the number of MCs in both the parietal and superior sagittal sinus (SSS) regions of the dura mater. CSL administration also did not result in significant differences in MC counts in either dura mater region. However, consistent with the LN results, CSL significantly decreased the degranulation rate of MCs in both regions compared to the LPS group. 

In the parietal region, the LPS group exhibited a higher degranulation rate (44.30 ± 12.33%) compared to the CON group (26.86 ± 7.26%, *p* < 0.01, Figure 5e). CSL administration reduced the degranulation rate in a dose-dependent manner, with a significant reduction observed only in the 30 mg/kg CSL + LPS group. The degranulation rates were 37.01 ± 6.28% for the 10 mg/kg CSL group, 32.55 ± 3.61% for the 20 mg/kg CSL + LPS group, and 30.14 ± 5.03% for the 30 mg/kg CSL + LPS group (*p* < 0.05). 

In the SSS region, the LPS group showed an increased degranulation rate (50.79 ± 8.36%) compared to the CON group (29.69 ± 6.03%, *p* < 0.001, Figure 5e). CSL administration reduced the degranulation rate in a dose-dependent manner, with significant reductions observed in all CSL + LPS groups, except the 10 mg/kg CSL + LPS group. The degranulation rates were 42.45 ± 4.92% for the 10 mg/kg CSL group, 36.90 ± 6.86% for the 20 mg/kg CSL + LPS group (*p* < 0.05), and 30.66 ± 7.76% for the 30 mg/kg CSL + LPS group (*p* < 0.001). Notably, the 30 mg/kg CSL + LPS group exhibited a significantly lower degranulation rate compared to the 10 mg/kg CSL + LPS group (*p* < 0.05, Figure 5e).

### 2.6. Inhibitory Effect of CSL on the Expression of TNF-α, and IL-1β in the PFC

The LPS group presented an increased TNF-α mRNA level, compared to the CON group (*p* < 0.001, Figure 6a). CSL administration reduced TNF-α mRNA levels in a dose-dependent manner compared to the LPS group, with a significant reduction observed in the 20 mg/kg CSL + LPS group (*p* < 0.01) and the 30 mg/kg CSL + LPS group (*p* < 0.01). The LPS group presented increased IL-1β mRNA level, compared to the CON group (*p* < 0.001, Figure 6b). CSL administration reduced IL-1β mRNA levels in a dose-dependent manner compared to the LPS group, while a significant reduction was observed only in the 30 mg/kg CSL + LPS group (*p* < 0.05).

## 3. Discussion

This study investigated the antidepressant-like effects of CSL in an LPS-induced depression model. The results from the SPT, TST, and FST indicated that CSL significantly reduced depressive-like behaviors. Furthermore, CSL notably modulated mast cell (MC) activation in the smLNs, dcLNs, and dura mater. Hematological analysis demonstrated that CSL effectively lowered the neutrophil-to-lymphocyte ratio (NLR), and analysis of the PFC revealed a significantly inhibited expression of pro-inflammatory cytokines like TNF-α and IL-1β.

Behavioral tests such as the SPT, TST, and FST are well-recognized tools for evaluating depression-like behaviors in preclinical models of depression. Anhedonia, characterized by a diminished capacity to experience pleasure, is a core symptom of depression and can be effectively modeled using the SPT, where a reduction in sucrose preference is indicative of anhedonia [26]. Conversely, the TST and FST are designed to assess behavioral despair, with increased immobility interpreted as a lack of escape-driven behavior, a feature commonly associated with depression [27,28]. LPS-induced depression models are widely used to replicate inflammation-related depressive conditions, as LPS administration elicits sickness behaviors, including decreased sucrose preference and increased immobility, by activating peripheral and central inflammatory responses. [29]. In our study, LPS administration led to decreased sucrose preference in the SPT and heightened immobility in both the TST and FST, aligning with its established depressive-like effects [18,30]. Treatment with CSL reversed these effects in a dose-dependent manner, restoring sucrose preference and reducing immobility, which suggests an antidepressant-like action. These results are consistent with previous research showing the efficacy of cannabinoids in mitigating inflammation-induced behavioral changes, likely through the modulation of neuroimmune pathways [14,18,31]. However, it is important to note that while a clear dose-dependent effect of CSL was observed in the FST and SPT, the TST did not demonstrate the same level of dose-dependency at the 30 mg/kg dose. This discrepancy may be attributed to the TST’s lower sensitivity to cannabinoid effects compared to the FST and SPT, which may engage distinct neurobiological pathways [17]. Additionally, pharmacokinetic factors, such as the distribution and metabolism of cannabinoids, could influence receptor activation differently across these tests [32]. Further investigations are warranted to explore these differences, with future studies incorporating additional dosing regimens and behavioral assays to more comprehensively assess the therapeutic potential of CSL.

The NLR is gaining recognition as a reliable indicator of systemic inflammation and is widely used in clinical and preclinical models to assess immune responses during stress conditions [33]. An elevated NLR reflects a disruption in immune regulation, primarily resulting from increased neutrophil activity and decreased lymphocyte levels, both indicative of heightened inflammatory responses [34]. Neutrophils function as the primary line of defense in the host immune response against pathogens, utilizing mechanisms such as chemotaxis, phagocytosis, and the release of reactive oxygen species (ROS), granular proteins, and cytokines [35]. As essential regulators of innate immunity, neutrophils recruit, activate, and modulate the functions of other immune cells through the secretion of pro-inflammatory and immunomodulatory cytokines and chemokines. This activity promotes the recruitment and activation of immune cells such as dendritic cells, natural killer (NK) cells, and CD4 and CD8 T cells [36]. In animal models, the administration of LPS is known to cause systemic inflammation, as evidenced by elevated NLR, indicating a peripheral inflammatory burden [37]. LPS activates various immune cells, including MCs and neutrophils [38,39]. Specifically, LPS binds to receptors on leukocytes, inducing the release of pro-inflammatory cytokines such as TNF-α, IL-6, and IL-1β [40]. NLR has been shown to change in MDD and has been associated with various factors, including chronic stress and impulsivity, both of which have been previously linked to suicidal behavior [41]. In our study, LPS administration significantly elevated NLR, indicating increased systemic inflammation. However, CSL treatment dose-dependently reduced NLR, suggesting a potential immunomodulatory effect. This reduction in NLR implies that CSL may help mitigate systemic inflammation, which plays a crucial role in inflammation-driven psychiatric disorders such as depression. Given that the endocannabinoid system regulates immune cell activity through cannabinoid receptors, it is plausible that CSL has an effect by inhibiting neutrophil activation and reducing cytokine production. While further research is needed, these findings suggest that CSL could be a promising therapeutic approach for managing inflammation-induced depressive disorders [42].

MCs are prevalent in nearly all vascularized tissues. Their close anatomical relationship with lymphatic vessels, combined with their capacity to synthesize, store, and release a wide range of inflammatory and vasoactive mediators, highlights their crucial role in regulating lymphatic vascular functions [43]. Recent studies suggest that MCs function as immune sentinels by presenting antigens through the expression of major histocompatibility complex II (MHC II) molecules. Additionally, they regulate the activity of various immune cells, including macrophages, dendritic cells (DCs), eosinophils, fibroblasts, and both T and B lymphocytes [44,45,46,47]. MC-derived histamine is essential for the LPS-induced phosphorylation of NF-κB [48]. LPS stimulates TLR4, initiating the histamine- and NF-κB-dependent production and secretion of various cytokines by MCs and surrounding tissue [48]. Upon encountering Gram-negative bacteria, MCs release inflammatory mediators, including TNFα, IL-6, and IL-1β. This response facilitates the chemotactic migration of NK cells, eosinophils, and neutrophils, while also enhancing the expression of adhesion molecules and chemokine ligands in lymphatic and blood vessels [49]. Additionally, MC-derived mediators affect lymphatic pumping by modulating the contractility of lymphatic muscle cells, which can either enhance or delay the transport of pathogens and immune cells to draining LNs [48,50]. In our study, the administration of CSL significantly inhibited excessive MC degranulation in both the smLNs and dcLNs, which are draining LNs connected to the meningeal lymphatic system [51,52]. Particularly, dcLNs are more directly connected to the meningeal lymphatic vessels and function as drainage LNs for macromolecules, cellular waste, toxic substances, and immune cells [52]. Aging is acknowledged as a significant risk factor for neurodegenerative disorders, including Alzheimer’s disease [53]. In aged mice, lymphatic drainage becomes compromised, resulting in a reduction diameter and coverage of meningeal lymphatic vessels, as well as diminished transport of CSF macromolecules to dcLNs [54]. Additionally, MC numbers in cervical lymph nodes are lower in young rats (2 months old) compared to other lymph nodes, but a more pronounced increase in MC density is observed in older rats (12 months old) [55]. Aging also leads to the basal activation of peri-lymphatic MCs, hindering the recruitment and activation of immune cells and potentially exacerbating inflammatory responses in these areas [56,57]. While this study was conducted on young healthy animals, the findings provide a baseline understanding of how CSL affects MC degranulation and neuroinflammation. The potential anti-inflammatory and neuroprotective effects observed in this context suggest that CSL may also have therapeutic relevance for age-related increases in MC activation and inflammation. Future studies should investigate whether these effects are preserved or enhanced in aging models, where chronic inflammation and impaired lymphatic function are more pronounced. MCs express both CB1 and CB2 cannabinoid receptors, and they exhibit diverse responses to cannabinoid exposure [58,59,60]. MCs also produce endocannabinoids, including an andamide, palmitoylethanolamide (PEA), suggesting the presence of a potential autocrine regulatory mechanism. Notably, CB1 possesses a unique ability to suppress FcεRI-induced secretory responses in MCs [61]. The inhibition of degranulation in dcLNs is consistent with our previous study on another species of *Cannabis sativa* L. [62], suggesting that CSL regulates MC activity in drainage LNs. Further research is needed to determine whether this regulation is associated with CSF drainage functions in the meningeal lymphatic system.

The vascularized meninges of the central nervous system (CNS) are essential for the protection, development, and maintenance of neural parenchyma [63]. The meninges are normal residents of innate immune cells, including macrophages, MCs, and innate lymphoid cells (ILCs). Additionally, circulating immune cells like neutrophils and T cells pass through the meninges as part of routine immunosurveillance [64,65,66]. Within the dura mater, the meningeal layer closest to the skull, MC distribution varies notably, with the highest density found in the interparietal region and the most pronounced activation occurring in the transverse region [67]. In humans, MCs in the convexity of the intracranial dura mater are situated near arterial and venous vessels, with a greater concentration observed around venous vessels within the periosteal layers and near the SSS [68]. In our study, MCs were more densely distributed in the parietal region compared to the SSS region, which is consistent with previous research [68]. No significant changes in MC number were observed following LPS or CSL administration. However, a notable increase in the degranulation ratio was observed in both regions, with the SSS region exhibiting a higher degranulation ratio after LPS administration. CSL treatment showed a dose-dependent suppression of degranulation in the SSS region. This suggests that the regulation of MC activation in the SSS region is functionally important, as this area is not only a venous site, but is also a critical region for the meningeal lymphatics. Since the discovery of meningeal lymphatics in 2015, an increasing body of evidence has highlighted the essential role of the meningeal lymphatics in regulating immune responses and inflammation within the central nervous system [52,54,69,70,71,72]. Moreover, recent studies suggest that the drainage of meningeal lymphatics, driven by VEGF-C, plays a crucial role in regulating depression-like behavior and enhancing the immune response against brain tumors [73,74,75]. MCs in the meninges may play a critical regulatory role in controlling brain inflammation and the progression of injury following a stroke [76]. Also, aging triggers the basal activation of perilymphatic MCs, which subsequently limits the recruitment and activation of various immune cell types within perilymphatic tissues [77]. CSL effectively suppressed the degranulation of MCs in the dura mater, particularly in the SSS region, suggesting that it plays a critical role in modulating mast cell activation in this area. These findings imply a complex interaction between mast cell activity and meningeal lymphatic drainage, which may serve as a crucial mechanism for regulating immune responses and inflammation in the central nervous system. Future studies should focus on elucidating the mechanisms by which CSL regulates meningeal lymphatic pathways through mast cell modulation, with particular emphasis on the VEGF-C-mediated pathway and its potential connections to neuroinflammatory or neurodegenerative diseases. Furthermore, it will be important to investigate how the basal activation of dural MCs during aging progresses and how CSL affects this activation. These studies could provide vital insights into developing preventive or therapeutic strategies for age-related neurological disorders and inflammatory diseases.

Pro-inflammatory cytokines such as TNF-α and IL-1β are pivotal in the neuroinflammatory response observed in LPS models, playing critical roles in the pathogenesis of neurodegenerative and psychiatric disorders, including depression and cognitive impairments. Our study demonstrates a significant increase in TNF-α and IL-1β expression in the PFC following LPS administration, consistent with previous reports in LPS models [18,29,30]. The administration of CSL markedly reduced these pro-inflammatory cytokines in a dose-dependent manner, particularly at the 30 mg/kg dose. This aligns with studies showing that cannabinoids like CBD and Δ9-THC, along with terpenes such as β-caryophyllene, exert potent anti-inflammatory and antidepressant effects in LPS-induced models by inhibiting pro-inflammatory cytokines through modulation of the NF-κB pathway and interactions with CB1 and CB2 receptors [78,79]. Further studies should focus on the precise mechanisms by which CSL exerts its anti-inflammatory and antidepressant effects. Investigating how CSL modulates the endocannabinoid system, including enzymes like FAAH and receptors such as GPR55, would provide key insights [80]. Additionally, exploring the roles of PPAR-γ, TRPV1, and the inhibition of NF-κB signaling in reducing cytokine expression and oxidative stress is needed [78,79,80]. 

Studies on the therapeutic potential of cannabis have predominantly focused on isolated cannabinoids, such as CBD and Δ9-THC, to explore their pharmacological properties. However, an increasing amount of evidence shows that treatments using the whole cannabis plant tend to produce more effective therapeutic outcomes compared to those relying solely on purified cannabinoids [81,82,83,84,85]. This superior efficacy is attributed to the “entourage effect,” a synergistic interaction between multiple bioactive compounds, including cannabinoids, terpenes, and flavonoids. In our study, we identified a variety of bioactive components in CSL, such as CBDA, CBCA, myrcene, and α-Pinene, which are likely to work together to produce the antidepressant-like effects observed. We propose that this synergistic interaction amplifies the therapeutic benefits, supporting the idea that whole plant formulations may offer superior efficacy for managing depression-like symptoms.

The limitations of our study include several important factors. First, we relied on the LPS-induced neuroinflammation model to assess the anti-inflammatory and antidepressant effects of CSL, which may not fully capture the complexity of depressive disorders in humans. Second, while CSL demonstrated significant effects on cytokine levels and behavior, the exact molecular pathways, particularly the interactions between cannabinoids and signaling mechanisms such as NF-κB and PPAR-γ, remain unclear. Furthermore, we did not assess the long-term effects or potential side effects of CSL, nor its efficacy in chronic depression models. Future studies should incorporate additional depression models and explore these mechanisms in greater depth to improve the translational relevance of our findings.

## 4. Materials and Methods

### 4.1. Animals

Male C57BL/6 mice (8 weeks old, 25–27 g) were obtained from Samtaco BIO (Gyeonggi-do, Republic of Korea) and housed in groups of 3–4 under a 12 h light–dark cycle with free access to food and water. All animal experiments were conducted in accordance with guidelines approved by the Wonkwang University Animal Experiment Ethics Committee (WKU24-64). 

### 4.2. Experimental Design

A graphical representation of the experimental procedure is shown in Figure 7. Mice were randomly assigned to five groups (*n* = 6 per group): (i) CON group (vehicle), (ii) LPS group (LPS only), (iii) 10 mg/kg CSL + LPS group (administered 10 mg/kg of CSL 30 min prior to 0.83 mg/kg LPS), (iv) 20 mg/kg CSL + LPS group (administered 20 mg/kg of CSL 30 min prior to 0.83 mg/kg LPS), and (v) 30 mg/kg CSL + LPS group (administered 30 mg/kg of CSL 30 min prior to 0.83 mg/kg LPS). Both CSL and LPS were administered i.p. at a dose of 10 µL/g of body weight. Mice that did not receive CSL or LPS were injected with the corresponding vehicle. Anesthesia was administered via intramuscular injection of an anesthetic cocktail containing Zoletil (20 mg/kg, Virbac, Carros, France) and xylazine (5 mg/kg, Bayer, Leverkusen, Germany). One week before the experiment, the mice were individually housed and habituated to a free choice of 1% sucrose solution and water. CSL was administered 30 min before the LPS injection. The TST and the FST were performed 24 h after LPS administration. SPT was assessed during that 24 h period. After the behavioral tests, blood, LNs, dura mater, and PFC samples were collected for the different analyses.

### 4.3. Drugs

*Cannabis sativa* L. inflorescence samples of the species ‘V1’ [86] were obtained from Nongboomind Company (Seoul, Republic of Korea) under the permission of the Ministry of Food and Drug Safety. *Cannabis sativa* L. was extracted with 70% ethanol. First, 300 mL of 70% ethanol was added to a flask containing 20.4 g of *Cannabis sativa* L. The *Cannabis sativa* L. soaked in 70% ethanol was ultrasonicated (frequency 40 kHz bath) at 40 °C for 30 min using an ultrasonicator (POWERSONIC 410, Hwashin Technology, Seoul, Republic of Korea). The extraction process was repeated three times. The extracted solution was then filtered through a 5 μM pore-size qualitative filter paper (ADVANTEC, Tokyo, Japan). The extract was concentrated under reduced pressure at 40 °C using a rotary vacuum evaporator (EYELA N-1110, Tokyo, Japan). Subsequently, the extract was freeze-dried at −80 °C and stored at −20 °C until use. A final 3.13 g dried powder was obtained from 20.4 g of *Cannabis sativa* L., and the final yield was 15.62%. For administration, the CSL was dissolved in a solution of 2% Tween 80 (Duksan, Seoul, Republic of Korea), 5% Propylene Glycol (Duksan, Seoul, Republic of Korea), and saline, and administered at doses of 10, 20, and 30 mg/kg [62]. LPS (Lipopolysaccharides from Escherichia coli O55:B5, Sigma-Aldrich, Darmstadt, Germany) was dissolved in saline and used at a dose of 0.83 mg/kg [29]. Identification and quantification of cannabinoids in CSL was analyzed using GC-MS, as previously reported [87]. The analyzed cannabinoids were a reference standard mixture of eight neutral cannabinoids (purity ≥ 99.0%) [cannabidivarin (CBDV), tetrahydrocannabivarin (THCV), cannabidiol (CBD), cannabinol (CBN), delta-9-tetrahydrocannabinol (Δ9-THC), delta-8-tetrahydrocannabinol (Δ8-THC), cannabichromene (CBC), cannabigerol (CBG)] and a standard mixture of 6 acidic cannabinoids (purity ≥ 98.5%) [cannabichromenic acid (CBCA), cannabidivarinic acid (CBDVA), cannabidiolic acid (CBDA), cannabigerolic acid (CBGA), tetrahydrocannabivarinic acid (THCVA), and tetrahydrocannabinolic acid-A (THCA-A)], and isotopically labeled internal standards such as Δ9-THC-d3, CBD-d3, CBDA-d3, and THCA-d3 (purity ≥ 99.9%). And HS-GC/MS was employed to identify the major terpenes in the CSL [86]. Quantitative analysis was performed on α- and β-pinene, myrcene, limonene, β-caryophyllene, α-humulene, etc., which are known as the major terpenes in *Cannabis sativa* L., in order to investigate terpenes.

### 4.4. Sucrose Preference Test (SPT)

This test was used to assess anhedonia. Prior to the experiment, animals were housed individually and given unrestricted access to both water and a 1% sucrose solution for one week to acclimate them to the availability of both liquids, as described in previous studies [26]. The positions of the two bottles were alternated daily to prevent any positional preference. Sucrose preference was measured 24 h after LPS injection by calculating the percentage of sucrose solution consumed relative to the total liquid intake during that time period [88].

### 4.5. Tail Suspension Test (TST)

Mice were suspended by the tail and video recorded for 6 min using a webcam (APC930U, ABKO, Seoul, Republic of Korea). Immobility time was defined as the period when the mice hung passively and remained completely motionless, indicating behavioral despair, following standard protocols [27].

### 4.6. Forced Swimming Test (FST)

The Forced Swim Test (FST) was performed according to established protocols [28]. The apparatus consisted of a transparent cylindrical glass container (15 cm in diameter, 20 cm deep) filled with water maintained at 23–25 °C, with a webcam (APC930U, ABKO, Seoul, Republic of Korea) positioned in front of the container for recording. The immobility time of the mice was recorded over a 6 min period, with the data analyzed during the last 4 min, excluding the initial 2 min for acclimatization. Immobility time was defined as the period when the mouse was floating with minimal movement, except those required to keep its nose above water, indicating behavioral despair.

### 4.7. Hematological Inflammatory Markers Analysis

In total, 150 µL of blood sample was collected from the heart following an abdominal incision and the careful removal of the ribs and diaphragm to ensure optimal access. This procedure was performed prior to perfusion for PFC sampling. The collected blood was immediately transferred into EDTA tubes (Greiner Bio-One, Chonburi, Thailand), kept refrigerated, and sent to the Korea Non-Clinical Technology Solution Center (KNTSC) for hematological analysis. The analysis focused on white blood cell differential counts, including neutrophils, lymphocytes, monocytes, eosinophils, and basophils. The NLR was calculated by dividing the neutrophil count by the lymphocyte count.

### 4.8. Staining of Mast Cells (MCs) in LNs

All procedures were performed under a stereomicroscope (SZX16, Olympus, Tokyo, Japan) for precise observation. Once deep anesthesia was confirmed, the mice’s legs were secured using tape to maintain positioning. The neck region was shaved with a razor blade to create a clear surgical area. A midline skin incision was made along the neck, allowing for access to the smLNs and dcLNs. The smLNs were the first to be identified and excised, as they are located superficially above the submandibular gland. Following this, the incision was deepened to access and collect the dcLNs, located deeper beneath the sternocleidomastoid muscle. Both the dcLNs and smLNs were fixed overnight in 10% neutral buffered formalin (NBF) for tissue preservation. The samples were subsequently dehydrated in a graded series of alcohols, cleared in xylene, and embedded in paraffin. Paraffin blocks were sectioned at 4 µm using a microtome and mounted onto slides for staining. MCs were stained using 0.5% toluidine blue in 0.5 N hydrochloric acid, as described in previous studies [89]. The LN area and MC count were evaluated using a phase contrast microscope (BX51, Olympus, Tokyo, Japan). The degranulation rate was calculated as the ratio of degranulated MCs to the total number of MCs in each LN.

### 4.9. Staining of Mast Cells (MCs) in Dura Mater

Following perfusion, the neck was severed, and the scalp was carefully incised to expose the skull. The skull, with the dura mater intact, was excised and fixed in 10% NBF for 24 h. To stain mast cells, the skulls were immersed in 0.5% acidic toluidine blue for 30 s and subsequently rinsed with phosphate-buffered saline (PBS, Gibco, Grand Island, USA) [67,90]. The dura mater was carefully separated from the skull, mounted onto glass slides, and coverslipped using mounting medium. For mast cell analysis, the dura mater was divided into two regions: the parietal and the SSS. In total, 8 non-overlapping 1 mm^2^ boxes were selected per region, and the average number of MCs was calculated. Quantification was performed using a phase contrast microscope (BX51, Olympus, Tokyo, Japan), and the degranulation rate was determined by calculating the ratio of degranulated MCs to the total number of MCs within each counting region.

### 4.10. Quantitative PCR (qPCR)

To prepare brain tissues for molecular analysis, the animals were perfused with cold PBS through the heart under deep anesthesia to remove circulating blood, ensuring clean tissue samples. After perfusion, the skull was carefully opened and the entire brain was removed. The prefrontal cortex (PFC) was then precisely dissected from the extracted brain. The isolated PFC tissues were immediately flash-frozen in liquid nitrogen and stored at −80 °C until RNA extraction. Total RNA was extracted from the PFC using the Easy-BlueTM RNA extraction kit (iNtRON Biotechnology, Sungnam, Republic of Korea). Reverse transcription of RNA to cDNA was performed using the ReverTra Ace qPCR RT Kit (Toyobo, Osaka, Japan). The ABI StepOne Plus detection system was used to perform TaqMan quantitative RT-PCR according to the manufacturer’s protocol. The housekeeping gene, β-actin, was used for normalizing the mRNA levels of the target genes. The PCR cycling conditions were as follows: 95 °C for 3 min, 60 cycles of 95 °C for 10 s, 60 °C for 10 s, and 72 °C for 20 s. The sequences of primers used for qPCR are shown in Table 1.

### 4.11. Statistical Analysis

All data are expressed as mean ± standard deviation (SD). Statistical analyses were performed using one-way analysis of variance (ANOVA) followed by Tukey’s post hoc test in SPSS for Windows (version 26.0; IBM Corp., Armonk, NY, USA). A *p*-value of <0.05 was considered statistically significant.

## 5. Conclusions

We demonstrated that CSL exhibits significant antidepressant-like and anti-inflammatory effects in an LPS-induced neuroinflammatory model. CSL administration effectively reduced depressive-like behaviors, as observed in the SPT, TST, and FST, and modulated the degranulation of MCs in LNs and the dura mater. Furthermore, CSL decreased the expression of pro-inflammatory cytokines TNF-α and IL-1β in the PFC in a dose-dependent manner. These findings suggest that CSL acts both through immune modulation and neuroinflammation suppression, possibly via the endocannabinoid system and pathways such as NF-κB, PPAR-γ, and VEGF-C. The synergistic interaction between cannabinoids and terpenes in CSL likely contributes to its therapeutic potential, supporting the notion of the “entourage effect”. While these results are promising, further studies are required to clarify the exact mechanisms involved and to assess the long-term safety and efficacy of CSL in chronic depression models.

## Figures and Tables

**Figure 1 pharmaceuticals-17-01409-f001:**
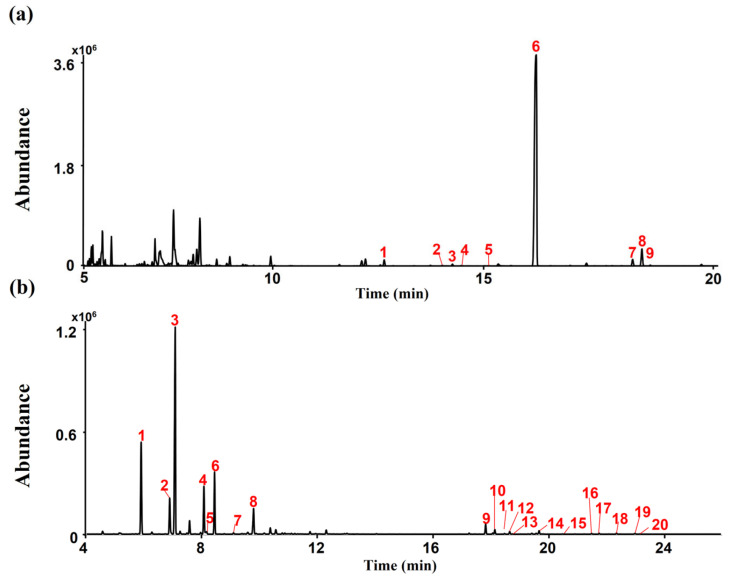
Total ion chromatograms (TICs) of CSL. (**a**) Peak identities by GC-MS are as follows: 1. CBD; 2. CBC; 3. Δ8-THC; 4. Δ9-THC; 5. CBG; 6. CBDA; 7. THCA; 8. CBCA; 9. CBGA. (**b**) Peak identities by headspace GC-MS are as follows: 1. α-Pinene; 2. β-Pinene; 3. Myrcene; 4. D-Limonene; 5. Eucalyptol; 6. β-Ocimene; 7. γ-Terpinene; 8. Sabienen hydrate; 9. β-Caryophyllene; 10. trans-α-Bergamotene; 11. E-β-Farnesene; 12. α-Humulene; 13. Alloaromadrene; 14. β-Bisabolene; 15. cis-α-Bisabolene; 16.Carophyllene oxid; 17 Guaiol; 18. γ-Eudesmol; 19. α-Eudesmol; 20.Bulnesol.

**Figure 2 pharmaceuticals-17-01409-f002:**
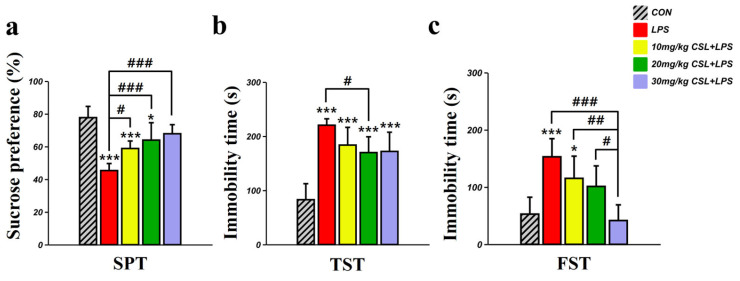
Behavioral effect of CSL administration in the LPS model. (**a**) Sucrose preference test, (**b**) immobility time in the tail suspension test (TST), and (**c**) immobility time in the forced swimming test (FST). Results are expressed as mean ± standard deviation (S.D.). One-way ANOVA followed by Tukey’s post hoc test. *, compared with the CON group; #, compared between the groups. */#, *p* < 0.05; ##, *p* < 0.01; ***/###, *p* < 0.001. (*n* = 6 per group).

**Figure 3 pharmaceuticals-17-01409-f003:**
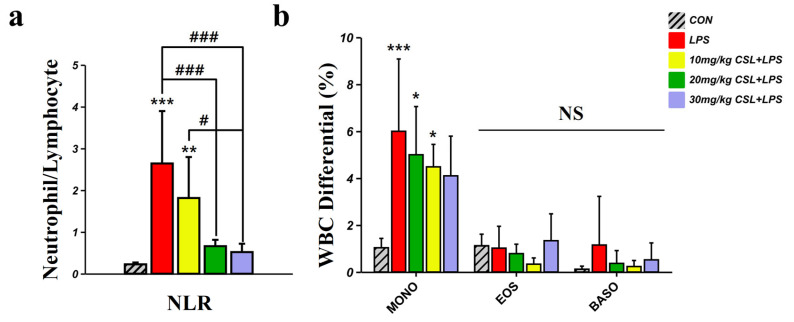
Hematological inflammatory markers following CSL administration in the LPS model. (**a**) Neutrophil to lymphocyte ratio (NLR), (**b**) monocytes (MONO), eosinophils (EOS), and basophils (BASO). Results are expressed as mean ± standard deviation (S.D.). One-way ANOVA followed by Tukey’s post hoc test. *, compared with the CON group; #, compared between the groups; NS, not significant. */#, *p* < 0.05; **, *p* < 0.01; ***/###, *p* < 0.001. (*n* = 6 per group).

**Figure 4 pharmaceuticals-17-01409-f004:**
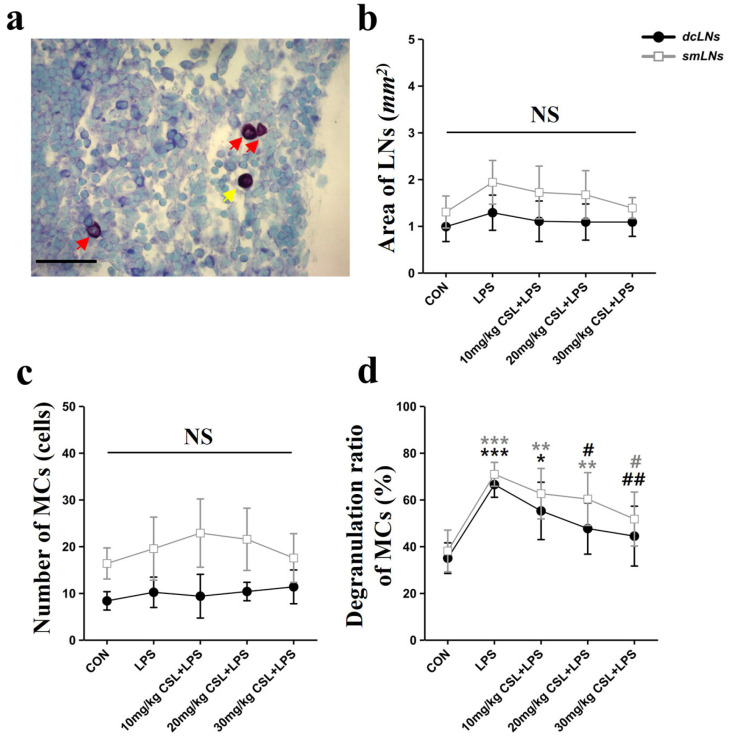
Effects of CSL on mast cell (MC) activation in deep cervical lymph nodes (dcLNs) and submandibular lymph nodes (smLNs) in the LPS model. (**a**) Representative LN image showing mast cell degranulation (red arrows: degranulated MCs, yellow arrow: non-degranulated MCs, scale bar: 100 µm), (**b**) area of LNs, (**c**) number of MCs, (**d**) degranulation ratio of MCs. Results are expressed as mean ± standard deviation (S.D.). One-way ANOVA followed by Tukey’s post hoc test. *, compared with the CON group; #, compared with the LPS group; NS, not significant. */#, *p* < 0.05; **/##, *p* < 0.01; ***, *p* < 0.001. (*n* = 6 per group).

**Figure 5 pharmaceuticals-17-01409-f005:**
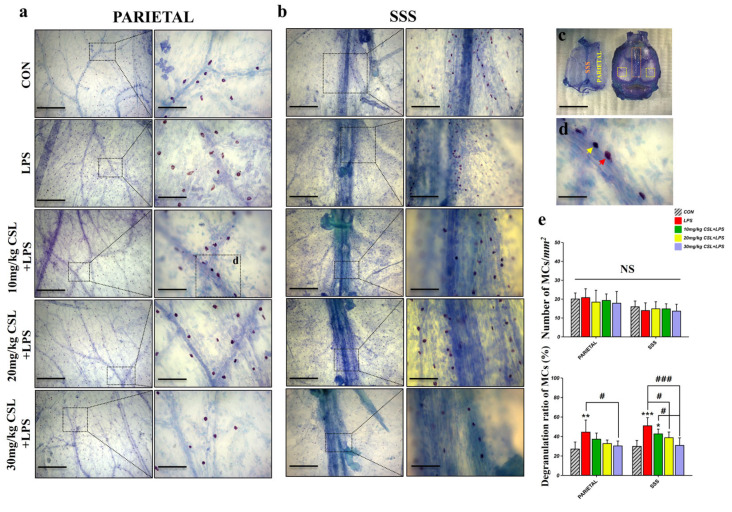
Effects of CSL on mast cell (MC) activation in the dura mater of the LPS model. (**a**) Representative images of MCs in the parietal region and (**b**) in the superior sagittal sinus (SSS) region. The right side of each image is a magnified view of the area highlighted by the dashed box. Scale bar: (**a**) left: 1 mm, right: 200 µm; (**b**) left: 1 mm, right (CON and LPS groups): 400 µm, (CSL groups): 200 µm. (**c**) Dura mater separated from the skull, with the parietal region marked by a yellow dashed box and the SSS region by an orange dashed box. Scale bar: 10 mm. (**d**) Representative image of MCs in the dura mater (red arrow: degranulated MCs, yellow arrow: non-degranulated MCs, scale bar: 100 µm). (**e**) Number of MCs/mm^2^, and degranulation ratio of MCs. Results are expressed as mean ± standard deviation (S.D.). One-way ANOVA followed by Tukey’s post hoc test. *, compared with the CON group; #, compared between the groups; NS, not significant. */#, *p* < 0.05; **, *p* < 0.01; ***/###, *p* < 0.001. (*n* = 6 per group).

**Figure 6 pharmaceuticals-17-01409-f006:**
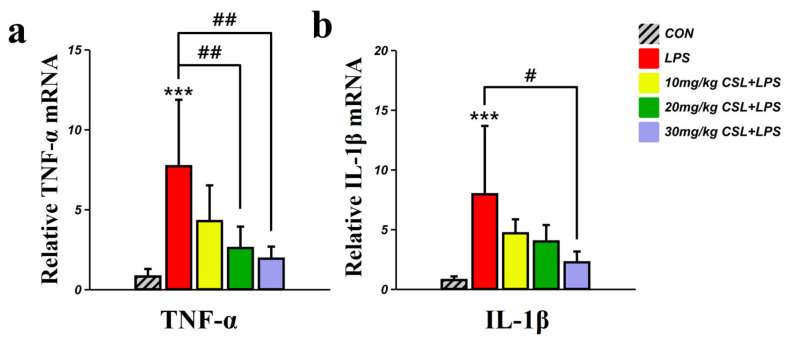
Inhibitory effects of CSL on the expression of pro-inflammatory cytokines in the prefrontal cortex (PFC) of the LPS model. (**a**) TNF-α mRNA level and (**b**) IL-1β mRNA level in the PFC. CSL administration significantly reduced the expression of TNF-α and IL-1β in a dose-dependent manner compared to the LPS group. Results are expressed as mean ± standard deviation (S.D.). One-way ANOVA followed by Tukey’s post hoc test. *, compared with the CON group; #, compared between the groups. #, *p* < 0.05; ##, *p* < 0.01; ***, *p* < 0.001. (*n* = 6 per group).

**Figure 7 pharmaceuticals-17-01409-f007:**
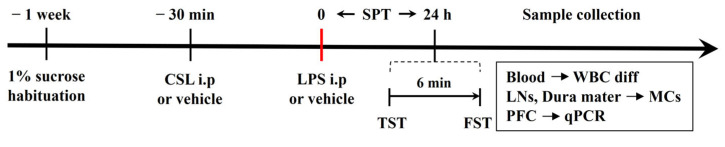
Experimental procedure of drug administration and behavioral tests performed in this study, and sample collection for the different analyses. CSL: *Cannabis sativa* L. inflorescence extract, LPS: lipopolysaccharide, SPT: sucrose preference test, TST: tail suspension test, FST: forced swimming test, WBC diff: white blood cell differential, LNs: lymph nodes, MCs: mast cells, PFC: prefrontal cortex, qPCR: quantitative PCR.

**Table 1 pharmaceuticals-17-01409-t001:** Sequences of primers used for qPCR.

Gene	Primer
TNF-α (F)	5′-TAC CTT GTC TAC TCC CAG GTT CTC-3′
TNF-α (R)	5′-GTG TGG GTG AGG AGC ACG TA-3′
IL-1β (F)	5′-ACC TGC TGG TGT GTG ACG TT-3′
IL-1β (R)	5′-TCG TTG CTT GGT TCT CCT TG-3′
β-actin (F)	5′-ATC ACT ATT GGC AAC GAG CG-3′
β-actin (R)	5′-TCA GCA ATG CCT GGG TAC AT-3′

## Data Availability

Data are contained within the article.

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
