# Peer review of "Antidepressant-like Effects of Cannabis sativa L. Extract in an Lipopolysaccharide Model: Modulation of Mast Cell Activation in Deep Cervical Lymph Nodes and Dura Mater"

_pharmaceuticals, 2024, doi:10.3390/ph17101409_

Round 1

Reviewer 1 Report

Comments and Suggestions for Authors

Summary:

The authors conducted research to investigate the antidepressant effects of Cannabis sativa L. extract in a lipopolysaccharide (LPS)-induced neuroinflammation model in mice. The study is timely and relevant, with clear data presentation through tables and figures to facilitate interpretation. The manuscript is written in quality scientific English, making it accessible to readers. Although the study is a valuable contribution to the field, some revisions are recommended to further enrich the manuscript for the Journal's audience. Below are major and minor comments for consideration.

Comments:

1.      Line 370: Please rephrase the sentence “MCs in the meninges as potential gatekeepers for modulating brain inflammation and pathology after a stroke” to improve clarity and precision.

2.      The authors acknowledge that a limitation of the study is the lack of investigation into the long-term effects of the extract. Since major depressive disorder is a chronic condition, why did the authors choose to administer a single dose of Cannabis sativa L. extract instead of a regimen with cumulative dosing over a longer period? Addressing this would provide a more relevant model for understanding the effects of the extract on depression.

3.      In most of the tests conducted, the Cannabis sativa L. extract produced effects in a dose-dependent manner, with the 30 mg/kg dose being particularly effective, except in the TTS. How do the authors explain this discrepancy? A discussion on possible reasons for this variance would strengthen the interpretation of the results.

4.      Regarding the results on mast cell (MC) degranulation: in the SSS area, the Cannabis sativa L. extract was observed to reduce degranulation in a dose-dependent manner, whereas in the parietal territory only the 30 mg/kg dose was effective, despite a higher density of MCs in this region. It would be beneficial to provide a statistical comparison of the effects between these two areas rather than presenting separate analyses. Such a comparison could offer deeper insights into regional differences in the extract’s action.

5.      In the Discussion, the authors delve into age-related neurological and circulatory diseases, although the current study was conducted on young, healthy animals. It is recommended that the authors either align the discussion with the experimental model or more thoroughly address the implications of their findings in aging models.

The manuscript is well-written, and the proposed revisions are aimed at enhancing its accuracy, depth, and relevance to the field.

Reviewer 2 Report

Comments and Suggestions for Authors

The manuscript by Joonyoung Shin at el., entitled “Antidepressant-Like Effects of Cannabis sativa L. Extract in the Lipopolysaccharide Model: Modulation of Mast Cell Activation in Deep Cervical Lymph Nodes and Dura Mater” reported that antidepressant- like and anti-inflammatory effects of Cannabis sativa L. extract in an LPS-induced neuroinflammation model, likely through modulation of cytokine expression and mast cell activity. The authors conclude that CSL as a potential therapeutic option for treating inflammation-related depression. The authors confirmed results with various techniques. The manuscript is well-written. However, some remarks should be taken by the authors into consideration.

Comments:

1.       The authors should include the schematic diagram experimental design. It will help to readers easily understand the study design.

2.       The authors should include the details for route of administration for the LPS and CSL extract in the Materials and Methods (MM) section.

3.       The authors should divide the subsection 4.1. into 4.2 as “Experimental design”.

4.       In Figure 2, the authors should represent the baseline values of Sucrose preference test as graph.

5.       For the Tail suspension test (TST) and Forced Swimming Test (FST), how was the immobility time was calculated? It was manually calculated or any software used? If software used to analyze, include the details in the MM section.

6.       In the MM section after the behavior analysis, the authors should include the detailed procedure for the animal anesthesia, blood and brain sample harvesting.

7.       In qPCR section, the authors should include the table for the primers.

8.       In statistical analysis section, the authors should use the mean ± standard error mean (SEM) instead of mean ± standard deviation (SD).

9.       Minor editing is needed throughout the manuscript, especially in the Discussion.

Round 2

Reviewer 1 Report

Comments and Suggestions for Authors

I would like to thank the authors for their detailed response to the review report. I accept the explanations provided and the revisions made to the manuscript. I now consider it suitable for publication.